# Flow Structure behind Spanwise Pin Array in Supersonic Flow

**Philip A. Lax [1], Skye Elliott [1], Stanislav Gordeyev [1], Matthew R. Kemnetz [2] and Sergey B. Leonov [1,*]**

1   Department of Aerospace and Mechanical Engineering, University of Notre Dame,
    Notre Dame, IN 46556, USA
2   Air Force Research Laboratory, Kirtland AFB, Albuquerque, NM 87108, USA
*   Correspondence: sleonov@nd.edu

**Abstract:** This work focused on the experimental characterization of a complex flow structure behind a cross-flow array of cylindrical pins installed on the wall of a supersonic duct. This geometry simulates several common gas dynamic configurations, such as a supersonic mixer, a turbulence-generating grid, or, to some extent, a grid fin. In this work, the instrumentation employed is essentially non-intrusive, including spanwise integrating techniques such as (1) fast schlieren visualization and (2) Shack–Hartmann wavefront sensors; and planar techniques, namely (3) acetone Mie scattering and (4) acetone planar laser-induced fluorescence. An analysis of the data acquired by these complementary methods allowed the reconstruction of a three-dimensional portrait of supersonic flow interactions with a discrete pin array, including the shock wave structure, forefront separation zone, shock-induced separation zone, shear layer, and the mixing zone behind the pins. The main objective of this activity was to use various visualization techniques to acquire essential details of a complex compressible flow in a wide range of temporal–spatial scales. Particularly, a fine structure in the supersonic shear layer generated by the pin tips was captured by a Mie scattering technique. Based on the available publications, such structures have not been previously identified or discussed. Another potential outcome of this work is that the details revealed could be utilized for adequate code validation in numerical simulations.

**Keywords:** cylindrical pin array; supersonic flow; fine flow structures; Mie scattering; PLIF

## 1. Introduction

The purpose of this work was to experimentally characterize a complex flow pattern behind an array of pins mounted cross-flow on the wall of a supersonic duct. In some ways, this geometry simulates various typical gas dynamic configurations, such as a supersonic mixer or a grid that creates turbulence. One other application is an excitation of well-controlled flow perturbations behind the pins array, replacing a stochastic flow pattern realized in a turbulent boundary layer or shear layer.

Shock wave boundary layer interaction (SWBLI) patterns produced by a circular cylinder installed on the wall of a supersonic duct have been widely studied, including the associated flow/BL structures [1–3], shock structures [4], and pressure distributions [5]. Experimental and numerical techniques have been used to explore the flow field around cylinders normal to the wall [6] and parallel to the wall [7]. Some scaling laws and correlations [8] have been found for simple geometries.

A single cylinder in a supersonic cross-flow is characterized by a bow shock in front of the cylinder and an unsteady subsonic wake immediately behind the cylinder. Further behind the cylinder, the flow is again accelerated to supersonic velocity. The flow field is characterized by a complex interaction between the shock waves and vortices generated behind the cylinder [9]. Near the solid boundary, the bow shock in front of a cylinder separates into a forward separation shock and a rear reattachment shock near the base of the cylinder, forming a classic lambda shape. The intersection of these shocks is referred to as the triple point. A separated flow region exists beneath the lambda shock. Just below the

triple point, a high-pressure supersonic jet penetrates the flow separation zone, creating two horseshoe vortices and a supersonic region immediately in front of the cylinder. At the upper end of the cylinder, the bow shock curves around the tip of the cylinder and continues propagating downstream. These characteristics are schematically illustrated in Figure 1a. The nature of the interaction is complex, leading to many features that are largely under-investigated but may be of great importance for the mixing efficiency.

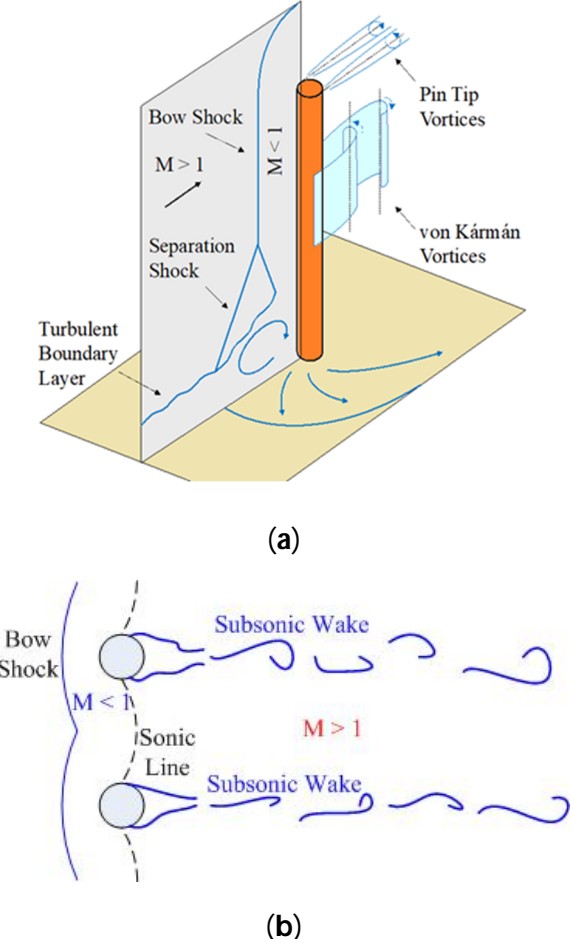

**(a)**

**(b)**

**Figure 1.** Flow schematics around (**a**) a single pin and (**b**) multiple spanwise pins in the cross flow.

Rather than a single cylinder, the current study is focused on a spanwise cylindrical pin array installed on a wall over a supersonic turbulent boundary layer. When multiple cylinders are placed in an array across the flow, the front bow shocks merge [10,11]. Flow immediately behind the combined bow shocks is subsonic, and a curved sonic line exists between the cylinders, past which the flow is again supersonic [12]. The subsonic wake and pin tip vortices continue propagating downstream and may interact with flow structures formed by adjacent pins. The resulting flow field is illustrated schematically in Figure 1b. Even though several relevant numerical works have been published, the three-dimensional detailed flow structure behind a pin array with free ends is largely under-investigated. A similar situation is the flow oscillation frequency induced by a pin–flow interaction. The vortex shedding oscillations behind an obstacle in incompressible flow are well studied [13,14], concluding that over a wide range of Reynolds numbers, $Re_d = 3 \cdot 10^2 - 10^5$ and the frequency of oscillation corresponds to $St_d = 0.18 - 0.22$, where $d$ is a characteristic dimension of the obstacle. For compressible supersonic flows, the published data [15,16], although rather limited, indicate the governing Strouhal number is about two times larger, $St_d = 0.4 - 0.45$. These data are related to a long cylinder installed cross-flow. In the case of the current study, the limited-height cylindrical pins are installed on the wall

and partially located within the BL, which makes the published data marginally applicable. The major objective of this work is to contribute to a basic understanding of the processes involved during pin array interactions with supersonic flow.

An analysis of the available publications indicates that the most relevant results have been acquired by numerical simulations. Despite significant progress in computational fluid dynamics (CFD), the simulation of a complex compressible flow is still challenging due to the approximate form of the specific gas equation of the state and transport coefficients, the discrete geometry of the meshes applied, and the stochastic nature of turbulence. Challenges in CFD include the prediction of transition and flow separation in viscous turbulent flows, model convergence in turbulent flows involving complex flow physics and/or complex geometries, mesh refinement, and uncertainty quantification, to list just a few unresolved problems [17,18].

In most cases, code validation is a predefined procedure. Ideally, the validation requires the consistency of numerical and experimental datasets in as wide a range of parameters as needed/possible. In practice, for a supersonic flow, this procedure is frequently reduced to a comparison to the calculated dataset of a wall pressure distribution taken with discrete pressure transducers and of a general flow structure (read: shock wave positions) visualized by a schlieren method. With such an approach, the details of the flow field can then be discussed based on simulated results alone, with many aspects remaining unvalidated. Presumably, a significant discrepancy might be observed, especially for mixing/shear layers, at SWBLI, vorticity-dominated flows, and other unsteady regions.

Proclaiming this, the secondary objective of this work is related to a potential contribution to a more detailed approach for the validation procedure using non-intrusive optical techniques to investigate and quantify difficult-to-observe features of a complex compressible flow. Targeting this objective, complementary to the schlieren visualization, three non-intrusive optical methods were applied to explore the details of a complex flow field consisting of a pin array interacting with a supersonic airflow: Shack–Hartmann wavefront sensing, acetone Mie scattering, and acetone planar laser-induced fluorescence (PLIF). These methods are sensitive to integrated gas density fields, number density/temperature composition, and gas number density, respectively. Mie scattering and PLIF are both local methods, excluding an integration over the flow depth.

PLIF is an optical method used to detect the population density of a chemical species present, or introduced to the flow field, through resonant coupling of the selected upper and lower energy levels of the target molecule. In chemically reacting flows, accessible tracer species are produced in reaction zones and heat-release areas. In non-reacting flows, a target molecule must be introduced. Acetone is frequently selected as a tracer species for PLIF measurements due to several favorable photophysical and thermodynamic properties [19–21]. These include a high partial pressure for simple seeding arrangements, adequate fluorescence quantum yield, low collisional quenching due to short fluorescence lifetimes, and low toxicity for safe user handling and bulk introduction to test arrangements. As the flammability limits of acetone [22] fall within the range of typical seeding concentrations, the application of an inert carrier gas is required in non-reacting test arrangements. In supersonic flow fields where low static temperatures are present, acetone condensation can affect the resulting PLIF signal, and a variation of stagnation temperature is required to mitigate condensation onset [23]. Acetone is frequently applied to study mixing processes in supersonic flows both locally in injected jets [23,24] and globally in the bulk flow [25]. Normally, the PLIF maps the gas density distribution. This work applies acetone PLIF in the bulk flow for planar visualization of shock structures at varying pin arrays.

Further details can be explored by Mie scattering performed on condensable media, assuming the condensation/nucleation is sensitive to not only the gas density but also to the gas temperature. The condensable vapors or gases are added to the high-pressure supply of the facility at a controlled partial pressure. As the static temperature of the gas decreases as it passes through the supersonic nozzle, the vapor condenses, forming small (nano/micro-scale), uniformly distributed droplets [26,27]. Since the droplets may

be smaller or larger than the wavelength of light used, the scattering of light is described by Mie scattering theory, which converges to the Rayleigh scattering approximation in the limit of nanoscale droplets. The droplets re-evaporate if the gas temperature is increased, making this method reasonably sensitive to temperature changes in the flow, such as those across shocks, mixing layers, and in flow separation zones.

Shack–Hartmann wavefront sensors are commonly used to measure the aero-optical distortions caused by the dependence of the air index of refraction on the gas density, as quantified by the optical path difference (OPD), in subsonic flow [28] atmospheric studies [29], as well as supersonic/hypersonic flows [30]. Since a Shack–Hartmann sensor measures the line-of-sight integral of fluctuations in the index of refraction caused by density inhomogeneity in the flow, it is used in this study to augment the qualitative schlieren imaging method in a horizontal plane, including a local flow velocity estimate, and provide an initial analysis of the flow perturbation pattern in the cross-flow direction, orthogonal to the pins-related shear layer.

In this manuscript, it is demonstrated how a combination of four non-intrusive optical methods reveals the detailed structure of a complex compressible flow, including the morphology of fine-scale vortical structures in the supersonic shear layer generated from the pin tips captured by a Mie scattering technique, a description of which is not present in the available literature.

## 2. Experimental Setup and Instrumentation

The SBR-50 facility at the University of Notre Dame is a supersonic blowdown wind tunnel with interchangeable Mach 2 and Mach 4 nozzles, with the Mach 2 nozzle being used here. An original feature of the facility is the compensation of the adiabatic gas cooling and pressure unsteadiness during the run time [31]. The test section has an upstream cross section of $76.2 \times 76.2$ mm with a $1°$ degree expansion on the top and bottom walls and a total length of 610 mm. Quartz side windows and fused silica upper and lower windows provide optical access to the test section. Stagnation conditions of the facility are $P_0 = 1$–4 bar and $T_0 = 300$–750 K provided by resistive heaters, with steady-state runtimes of up to 1 s. Working fluids include filtered, dried air as well as pure $N_2$ and gas mixtures. A total of 48 static pressure ports are distributed along the upper and lower test section walls, and a 16-probe Pitot rake may be installed at the end of the test section. Standard instrumentation of the facility includes a 64-channel pressure scanner (Scanivalve MPS4264) with an 800 Hz acquisition rate and a high-speed schlieren system.

### 2.1. Test Geometry

The interchangeable pin array inserts are composed of seven stainless steel cylindrical pins press-fit into threaded aluminum inserts, one of which is installed on the upper wall of the test section. The pin array inserts have a pin diameter $d = 2.5$ mm, pin spacing $l = 10$ mm center-to-center, and three different pin heights, $h = 25$ mm, 10 mm, and 5 mm. The thickness of the incoming turbulent boundary layer was estimated to be $\delta = 4 \pm 0.5$ mm. In addition to quartz panoramic side walls, rectangular fused silica windows 2 inches in the spanwise direction and 4 inches in the streamwise direction are flush–mounted on both the upper and lower test section walls downstream of the pin array inserts to provide optical access to the wake region. An illustration of the test section is shown in Figure 2. For the PLIF and Mie scattering visualization, the laser sheet was arranged in two ways: in a flow-wise direction (XY plane) and a cross-flow direction (YZ plane). The laser sheet positions are variable on the X and Z axes, correspondingly.

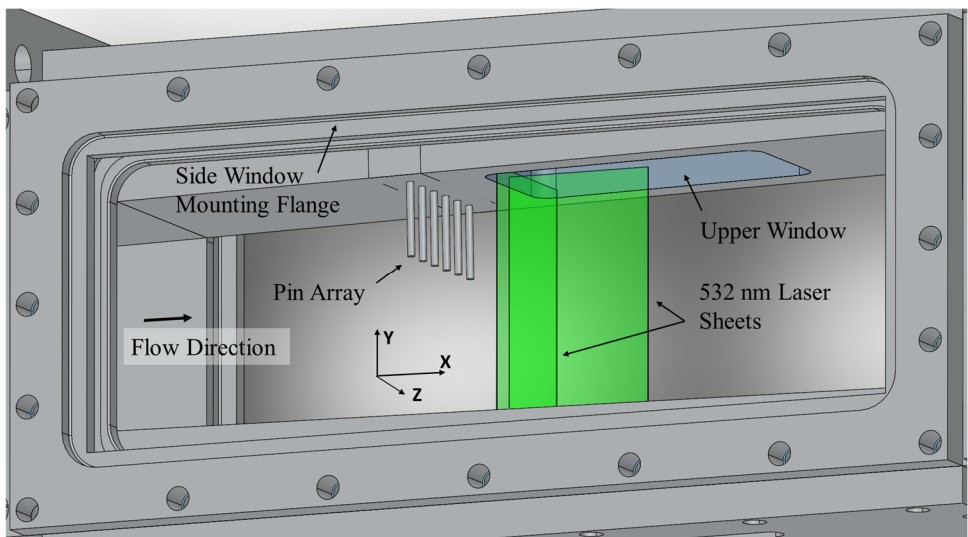

**Figure 2.** Illustration of the test section arrangement.

### 2.2. Schlieren Technique

The schlieren system is built on conventional refractor-based schematics. The light source consists of a high-current broadband white LED (Luminus™ Devices CFT-90-WCS-X11-VB600) powered by a pulsed LED driver (PicoLAS™ GmbH LDP-V 240-100 V3.3) with a pulse width of about 0.2 μs with an LED current of 80 A. The light source is focused using a 50 mm f = 32 mm aspheric condenser lens (Thorlabs™ ACL50832U-A) and a 50 mm f = 350 mm achromatic doublet (Edmund Optics™ 49-289-INK) and collimated and refocused using two 120 mm f/8.33 refractors. Imaging is performed with a high-speed CMOS camera (Vision Research Phantom™ v2512) fitted with a Nikon 200 mm f/4 AI-s lens. Two schlieren techniques are utilized: a traditional knife-edge method and a central dot method. The knife-edge method is sensitive to density gradients in one direction, while the central dot method, performed by placing a small black dot on a fused silica plate, is sensitive to density gradients in two directions. Since the field of view is limited to 120 mm, schlieren images of the test section length are composed by stitching together several images taken at different positions of the collimated optics with respect to the test section.

### 2.3. Acetone PLIF and Mie Scattering

Bulk seeding with acetone of the SBR-50 working gas is performed by adding pressurized liquid acetone to a pure $N_2$ stream via a liquid atomizer. The mixture is first passed through a static mixer with internal baffles and then into a small cell with sapphire windows. Acetone concentration is measured by UV absorbance using a 2 mW 280 nm UV LED with collimation and focusing optics and a GaP photodiode combined with a pressure transducer and a resistance temperature detector (RTD). Acetone is seeded at 2–6% by volume.

Laser excitation is performed using the 4th harmonic of an Nd:YAG laser (Quanta-Ray DCR-4) with a 10 ns pulse width and 10 Hz repetition rate. After frequency doubling from 1064 nm to 532 nm and again from 532 nm to 266 nm, the pulse energy is approximately 70 mJ/pulse. Sheet-forming optics are then used to produce a 25 mm wide 266 nm laser sheet that is approximately 200 μm thick at the sheet waist within the test section. The laser sheet forming arrangement is illustrated in Figure 3.

In many circumstances, the PLIF method allows quantitative measurements of the gas density or, at least, a semi-quantitative interpretation of the data acquired since, a priori, the luminescence signal is linearly proportional to the seeding gas/vapors concentration. In the case of acetone seeding, one of the issues is the acetone condensation onset, which is overcome in this work by proper adjustment of the gas temperature [23]. At the same time,

for the PLIF method with acetone seeding, it is challenging to properly estimate the gas density measurement errors.

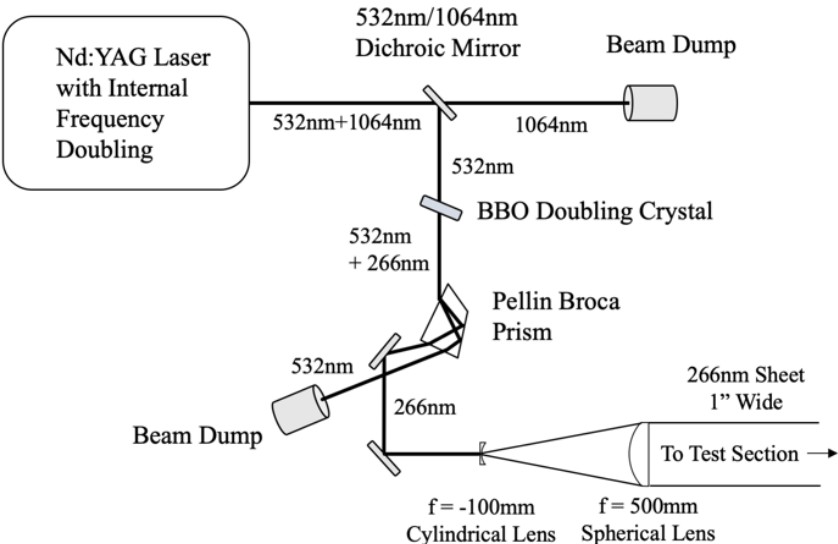

**Figure 3.** Laser sheet forming schematics.

For the Mie scattering visualization, the flow is seeded by nucleating any acetone vapors that occurred by lowering the static temperature of the flow until the acetone condenses, forming acetone nano/micro-droplets. Since the intensity of Mie scattering is much greater than the intensity of acetone PLIF, laser sheet forming for Mie scattering omits the second stage of the frequency doubling and uses an f = −50 mm cylindrical lens instead of an f = −100 mm lens to form a 50 mm × 532 nm sheet rather than the 25 mm × 266 nm sheet illustrated in Figure 3. In this work, the Mie scattering technique is considered to be rather qualitative, allowing the visualization of fine structures in the flow that are difficult to recognize by other methods.

*2.4. Optical Setup Using a Shack–Hartmann Wavefront Sensor*

Another way to study the density field and the dynamics of the flow perturbations is to collect spatially–temporally resolved wavefront data, as the wavefront's position is proportional to the line integrals of the gas density [28,32]. A schematic of the optical set-up is presented in Figure 4a. To collect the wavefronts downstream of the pin arrays, the 532 nm laser beam from a CW Nd:YAG laser was first expanded to 25 mm using a beam expander and then passed through a beam splitter cube. From the beam splitter, the beam was further expanded to 50 mm using a pair of lenses. Using steering mirrors, the beam was sent through the test section 50 mm downstream of the pin array either in the wall-normal direction through the top and the bottom optical inserts, indicated by a blue box in Figure 4b, or along the spanwise direction through the test section windows, indicated by a green circle in Figure 4b. After passing through the test section, the beam was reflected back along the same optical path using a large, flat return mirror. This so-called double path setup doubles the optical signal. The beam splitter cube reflects the return laser beam sideways and, after passing through a pair of re-imaging lenses, the beam is sent to the Shack–Hartmann wavefront sensor (SHWFS). The sensor consists of a Vision Research Phantom v1611 high-speed camera with a mounted lenslet array. The lenslet array has a pitch of 0.3 mm and a focal length of 38.2 mm. Time-resolved two-dimensional wavefronts were collected at two different sampling frequencies and two spatial resolutions. The first data set was collected at a lower sampling frequency of 39 kHz with a spatial resolution of 62 × 64 subapertures. The second data set was collected at a higher frequency of 99 kHz with a reduced spatial resolution of 34 × 38 subapertures.

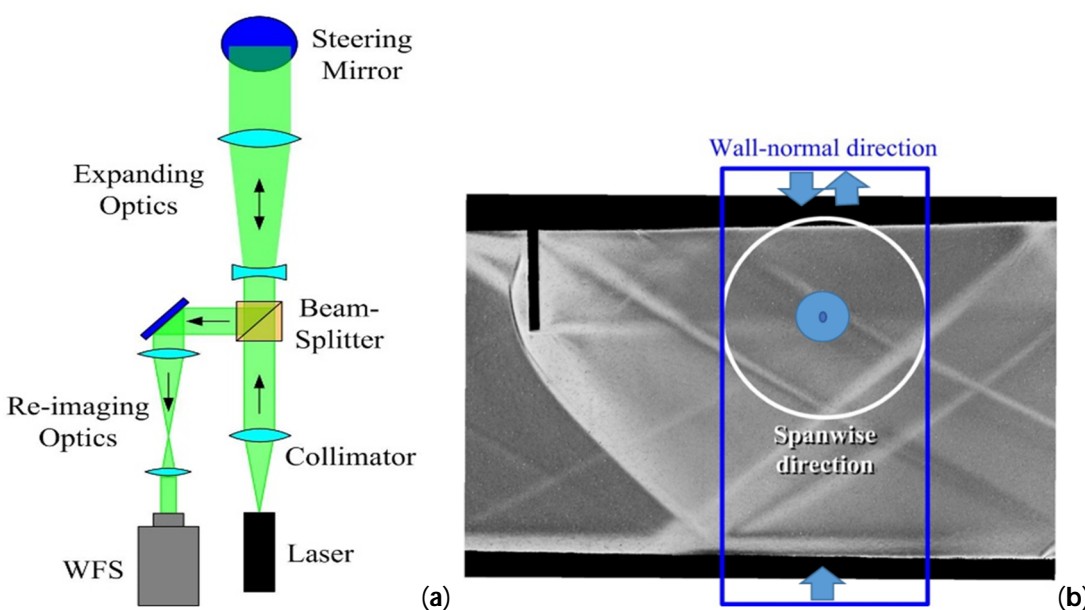

**Figure 4.** (**a**) Schematic of the optical set-up using a Shack–Hartmann wavefront sensor (WFS) and (**b**) laser beam λ = 532 nm arrangements in the wall-normal (blue box) and in the spanwise (white circle) directions.

The Shack–Hartmann wavefront sensor measures the local deflection angles, which are spatial gradients of the wavefronts. A Southwell method [33] was implemented to reconstruct the time-resolved wavefronts, $OPD(x, y, t)$, from the deflection angles using in-house software. After removing the instantaneous piston and tip/tilt modes from each wavefront, the spatial maps of $OPD_{rms}(x, y)$, defined as:

$$OPD_{rms}(x, y) = \sqrt{\overline{OPD^2(x, y, t)}}$$

were computed for each dataset. Here, an overbar denotes time averaging.

For the data set along the spanwise direction, dispersion analysis was also used to estimate the convective velocities at different distances from the wall. In [30,34], the dispersion analysis of the deflection angles was demonstrated to be effective in characterizing convecting optical disturbances. The dispersion analysis uses time-resolved streamwise deflection angles in several consecutive streamwise points, $\theta_x(x, t)$, to numerically compute a two-dimensional autospectral density function of the deflection angles, $S_\theta(f, k_x)$, using the standard procedure [35]. A stacking approach [30] was used to de-alias the spectra in the frequency domain. If the flow structures convect with a constant speed, the spectrum will have a single branch with a fixed slope. The convecting velocity, $U_C$, both the direction and magnitude, can be computed from the branch slope as $U_C = 2\pi f / k_x$.

## 3. Results and Discussion

### 3.1. Overall Flow Structure

In general, the shock structure near a wall-normal circular cylinder is composed of a forward separation shock (1), which joins a normal bow shock (2) and a rearward shock (3) closer to the base of the cylinder at the triple point to form the classic lambda shock structure. This structure is clearly visible in the experimental composite schlieren images in Figure 5, with a central dot optical arrangement in this case. A subsonic separation region (4) is formed beneath the lambda shock base, while the bow shock (5) curves around the tip of the cylindrical pin to form a nearly straight oblique shock (6) that reflects off the opposite wall. An oblique shock (7) is formed near the downstream base of the pin by the reattachment of the boundary layer. A shear layer (8) is formed at the pin tips and

propagates downstream. An upcoming boundary layer (9) is well visible on the top and bottom walls, where the large eddies are recognizable.

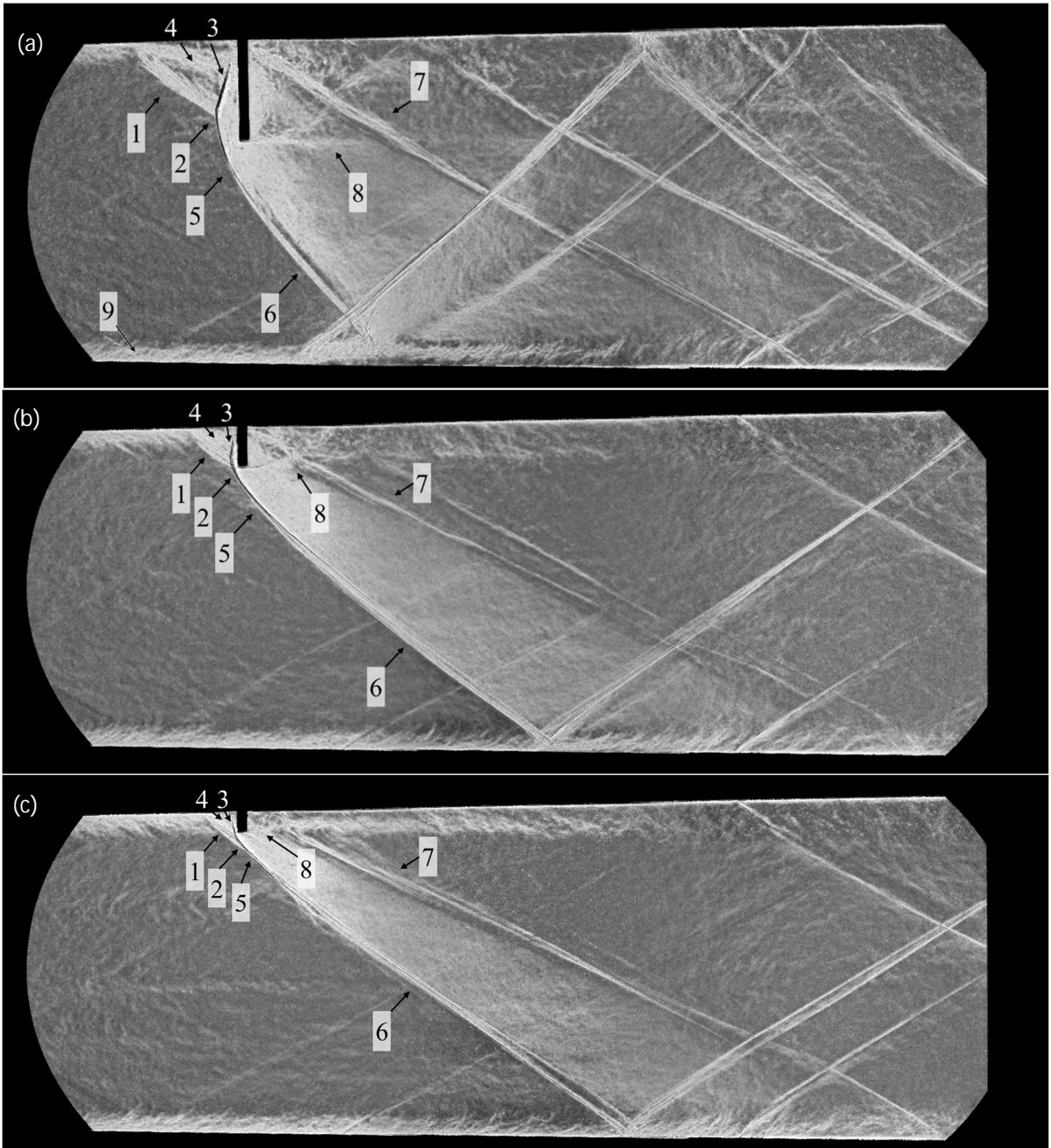

**Figure 5.** Instantaneous composite schlieren images of (**a**) 25 mm pins, (**b**) 10 mm pins, and (**c**) 5 mm pins.

Detailed views of the 25 mm pin array are given in the schlieren images of Figure 6, which uses vertical knife edge arrangements. Visible in these images are the lambda shock structure upstream of the pin array (1); a bow shock (2), which curves around the pin tips; oblique shock impingement on the boundary layer of the bottom wall with boundary layer separation (3); reflected oblique shock (4), expansion fan (5), and reattachment shock (6); and an expansion fan at the pin tips (7), an oblique shock originating near the pin base due to boundary layer reattachment (8), flow acceleration through the pin array (9), and an oblique shock crossing the supersonic region downstream of the pin array (10). Weak Mach

waves (11) are present upstream of the bow shock originating from small imperfections on the walls.

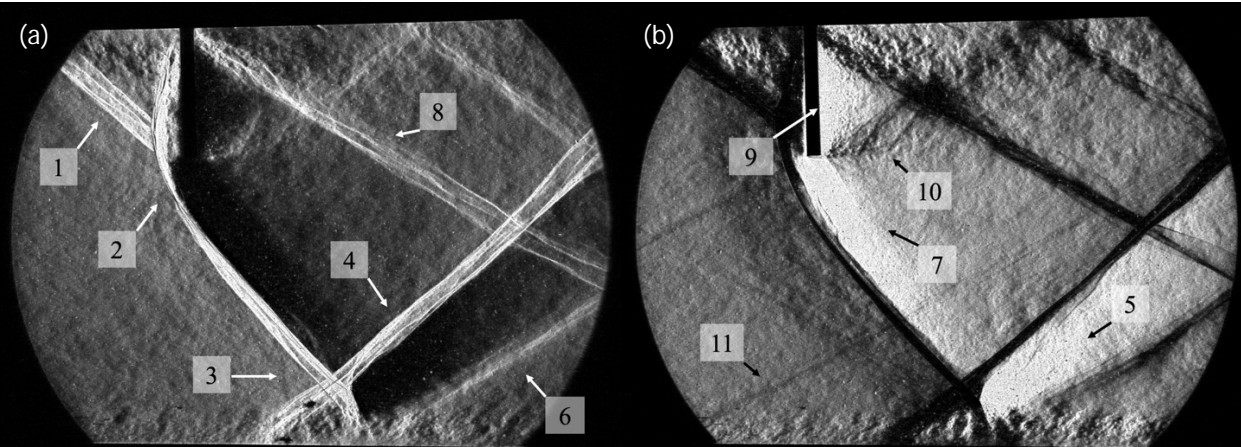

**Figure 6.** Instantaneous schlieren with a vertical knife edge with the vertical knife on the left (**a**) and right (**b**) sides.

The Mach number in the 2D nozzle formed by the pin array can be estimated by considering an array of cylinders of infinite length. The Mach–area relation [36] for $\gamma = 1.4$ flow is:

$$\frac{A_*}{A} = \left(\frac{\gamma+1}{2}\right)^{\frac{\gamma+1}{2(\gamma-1)}} M \left(1 + \frac{\gamma-1}{2}M^2\right)^{-\frac{\gamma+1}{2(\gamma-1)}} = \frac{216}{125}M\left(1 + \frac{M^2}{5}\right)^{-3},$$

where '$*$' indicates sonic conditions. Using the pin dimensions given in Section 2.1, the area fraction of the 2D nozzle formed by adjacent pins is $A_*/A = 3/4$, and the two solutions to the Mach–area relation are $M = 0.50$ and $M = 1.70$. Using normal shock equations and a nominal upstream freestream Mach number of $M = 2$ and assuming 2D planar flow, the flow immediately downstream of the normal shock in front of the pin array is subsonic with a Mach number of approximately $M = 0.58 > 0.50$, indicating that the flow between the pins is sonic and the flow downstream of the pin array is supersonic with a Mach number of approximately $M = 1.7$. However, since the pins have a finite height, these values should only be considered an estimate of the actual flow Mach number.

From the oblique shock relations, the flow downstream of an oblique shock in Mach 2 flow is sonic at an oblique shock angle of $61.5°$. By analyzing the schlieren images of Figure 6, it is found that flow immediately downstream of the bow shock curving around the pin array is subsonic for $y \leq 9$ mm from the end of the pins in the wall-normal direction. This subsonic flow is reaccelerated to supersonic speed by the expansion fan at the end of the pin tips.

A weak shock wave enters the flow field in the bottom left corner of Figure 6b, which is created by a small step on the bottom wall. Since this step is 25–50 µm high, this weak shock may be considered a Mach wave with infinitesimal strength. The wave angle of this Mach wave downstream of the oblique shock formed by the bow shock around the pin array and the expansion fan emanating from the pin tips is about $\theta = 29°$ with respect to the horizontal. Since the flow is reaccelerated by the expansion fan emanating from the pin tips immediately downstream of the oblique shock, the streamline direction downstream of the expansion fan is approximately horizontal. Using the relation $\sin \theta = 1/M$, it is found that the Mach number is about $M = 2$, indicating that after the expansion fan emanated from the pin tips, the flow behind the straight oblique shock returned to approximately the same Mach number as the flow in the undisturbed upstream region. However, due to the greater shock strength in the curved bow shock region near the pin array, the Mach number in this region is $M < 2$.

The wave angle of the oblique shock originating from the pin tips and crossing the accelerated flow region directly behind the pin array is about $\theta = 39°$, indicating that the Mach number in this region is $M \geq 1.6$, which is in agreement with the above estimate of $M = 1.7$. However, since this oblique shock has finite strength and the turning angle is not known, the Mach number in this region cannot be precisely calculated.

### 3.2. PLIF

PLIF images were taken using a 25 mm wide x 266 nm thick sheet positioned on the centerline of the test section. Stagnation conditions are $P_0 = 2.6$ bar and $T_0 = 425$ K with 6% acetone by volume in $N_2$. The upstream edge of the laser sheet is positioned against the upstream edge of the optical window and 25 mm from the pin arrays, which are in the upper left-hand side of the images in Figure 7. Since the pin arrays contain an odd number of pins, the laser sheet is positioned directly behind the middle pin.

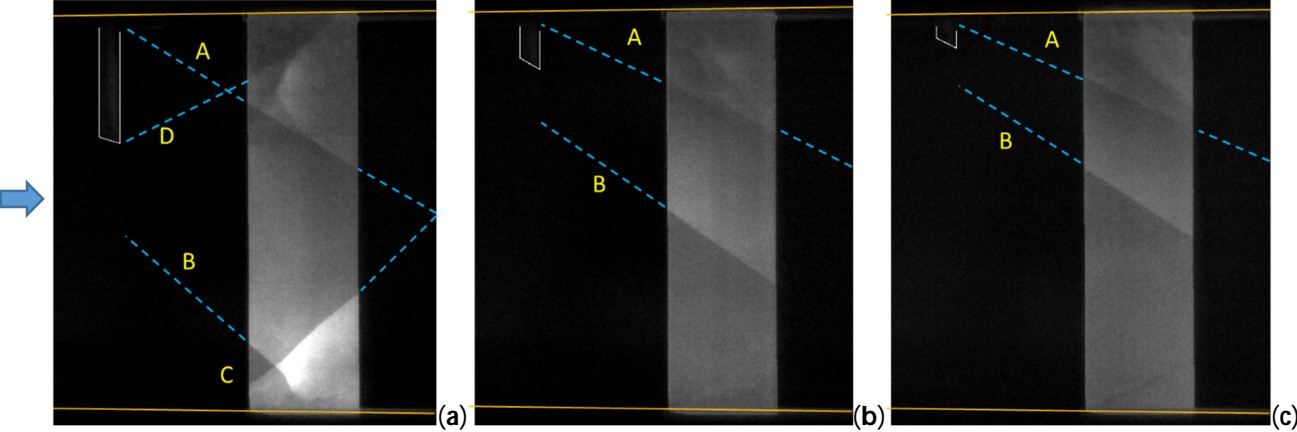

**Figure 7.** Single shot PLIF images behind (**a**) 25 mm pins, (**b**) 10 mm pins, and (**c**) 5 mm pins. Pin array outlined in white.

An oblique shock "A" originating near the pin base due to boundary layer reattachment is visible in the upper portion of Figure 7a. The strong oblique shock "B" from the pin array impinges the BL near the bottom wall in Figure 7a, providing a SWBLI pattern with a separation bubble, and reflected oblique shock "C" is especially evident. Since the acetone fluorescence intensity is a linear function primarily of flow density, brighter regions in the PLIF images downstream of the shock waves indicate regions in the flow with higher density. The density of the flow region in Figure 7a is greatest immediately downstream of the reflected oblique shock. The shock-induced separated area near the wall is also visible in Figure 7a. The corner shock "A" and oblique shock "B" are visible in Figures 5b and 7a,c, but progressively less so due to the weakening of the shocks with the decreased height of the pins.

The quantitative density fields presented in Figure 8 are recalculated from the PLIF images by linearly extrapolating the intensity of the laser sheet image after subtracting the off-plane background to account for the sensor dark current and background scattering, and scaling by the average pixel value in the freestream region. The density of the freestream $\rho_\infty = 0.7$ kg/m$^3$ is obtained from isentropic relations with a nominal freestream Mach number of $M = 2$.

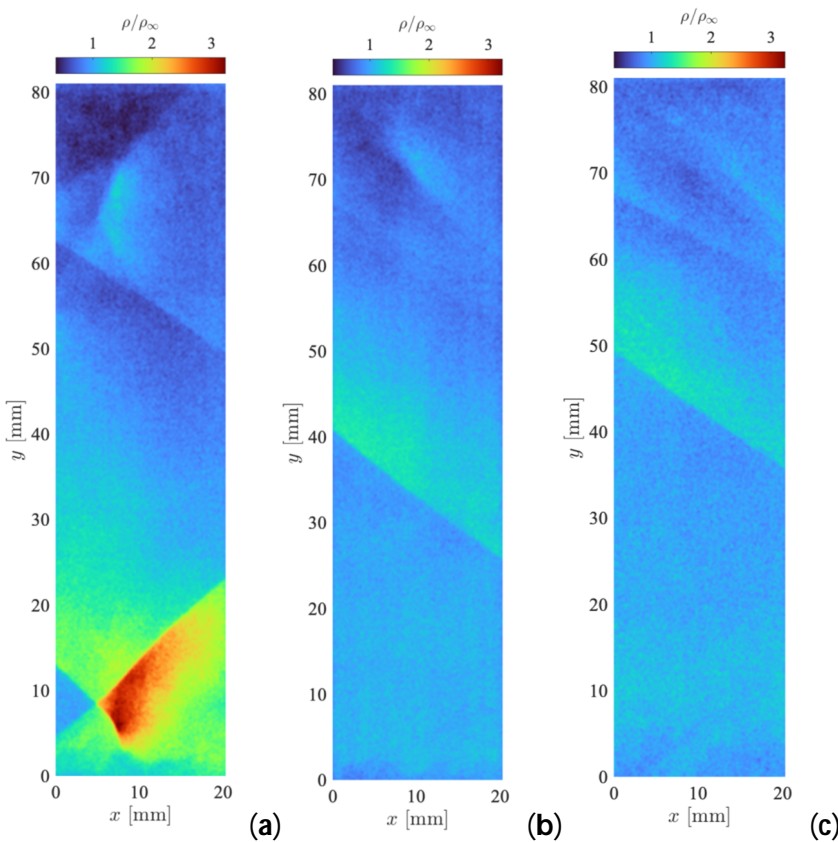

**Figure 8.** Gas density fields calculated from instantaneous PLIF images at (**a**) 25 mm pins, (**b**) 10 mm pins, and (**c**) 5 mm pins.

The density ratio for an oblique shock with angle $\theta$ and upstream Mach number $M$ for $\gamma = 1.4$ flow is:

$$\frac{\rho}{\rho_\infty} = \frac{6M^2 \sin^2 \theta}{M^2 \sin^2 \theta + 5}.$$

The oblique shock "B" entering the bottom left of Figure 7a has an angle of about $\theta = 45°$, and the resulting density ratio of $\rho/\rho_\infty = 1.7$ is in good agreement with Figure 8a. The gas compression area near the bottom wall is formed as a result of the impinging "B" and the reflected "C" shock interaction. The measured gas density is about $\rho/\rho_\infty = 2.6$, which is close to the critical value $\rho/\rho_\infty = 2.7$ for the formation of an irregular reflection with the Mach stem. Another flow structure is realized behind the pins array and is characterized by the presence of rarefaction and compression zones. In the rarefaction area, the gas density ratio varied from about $\rho/\rho_\infty = 0.5$ for h = 25 mm pins up to a negligible value for h = 5 mm pins. The compression zone with up to $\rho/\rho_\infty = 1.5$ for h = 25 mm pins is the result of a 3D interaction of the corner shock "A" and a relatively weak conical shock "D" coming from the pins' tips. The rarefaction and compression zones behind the pins array are recognizable for h = 10 mm pins and significantly faded for h = 5 mm pins.

*3.3. Wavefront Measurements Using the Shack–Hartmann Wavefront Sensor (SHWFS)*

Figure 9 shows the spatial map of $OPD_{rms}$ in the wall-normal direction for the pin array with 25 mm height. For reference, the spatial map is plotted relative to the locations of the tunnel side walls and the pin array. One main feature of the spatial map is two thin regions of increased optical distortions, related to shocks, crossing roughly in the middle of the aperture. By tracing back these shocks, it was concluded that these shocks are reflected bow shocks, originating from the first and the last pins. The angle of these shocks is approximately 30 degrees relative to the flow direction, suggesting a wall-normal

average Mach number of about $M = 2$. Individual wavefronts were also analyzed in an attempt to detect any regular aero-optical distortions due to Von Karman vortices shed from the pins. Unfortunately, by the time they reached the laser beam, the spanwise nature of the vortices was most likely sufficiently distorted and unable to form any discernable spatially regular optical distortions.

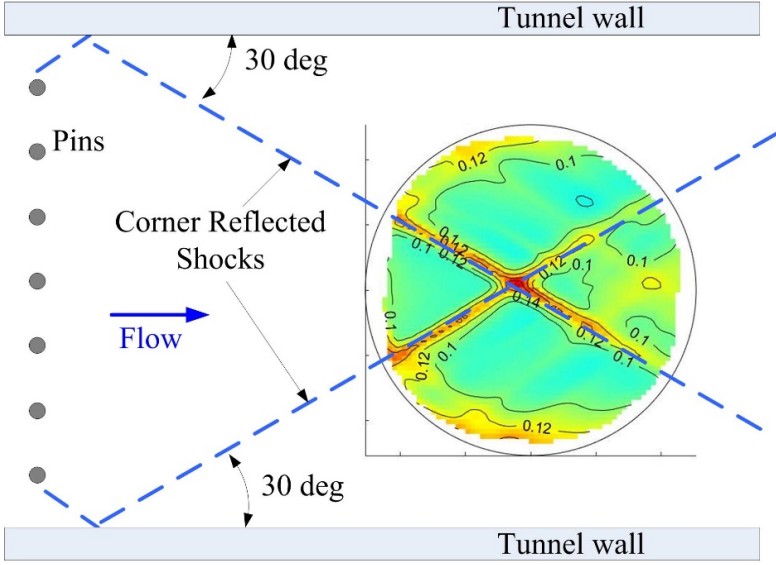

**Figure 9.** The spatial map of OPDrms (in microns) in the wall-normal direction, relative to the pins' positions and tunnel walls. The pin height is 25 mm.

Figure 10 presents the spatial map for the spanwise measurements for 25 mm pins, overlaid with the time-averaged schlieren image. A series of shocks can be clearly observed both in the middle and near the perimeter of the aperture. These shocks can also be found in Figure 6. The region y < 10 mm corresponds to the energized boundary layer at the wall, with the associated increase in density fluctuations. The weak shear layer at approximately y = 25 mm, originating from the pin tips, can also be observed.

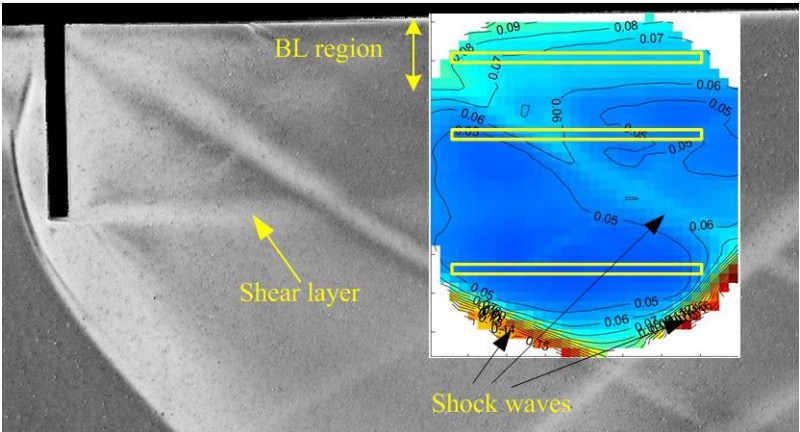

**Figure 10.** The spatial map of OPDrms (in microns) in the spanwise direction, overlaid with the time-averaged schlieren image. Locations to perform the dispersion analysis are indicated by yellow boxes. The pin height is 25 mm.

Three different wall-normal locations were chosen to perform the dispersion analysis and estimate the convective speeds at these locations. One location is y = 33.6 mm, which is in the freestream region, well above the pins. The second location is at y = 15.6 mm, which is in the wake region downstream of the pin array. The third location was chosen

to be inside the wall boundary layer at y = 4.8 mm. Figure 11 presents the stacked auto-spectral density functions for these three different wall-normal locations. At the location of y = 33.6 mm, which is above the wake introduced by the pin array, the spectrum (see Figure 11a) exhibits predominantly one branch, indicated by a solid black line, moving supersonically at approximately the freestream speed. In the wake region, see Figure 11b, this branch is also present. In addition, the spectrum reveals the presence of a range of subsonic convective speeds, with a maximum speed of approximately 0.4 of the freestream, denoted by the dashed red line. Thus, two types of regions exist in the pin wake region. One of them travels at a supersonic speed, close to the freestream speed, and represents regions between the pins. Another type has a range of slower speeds between 0 and 0.4 $U_\infty$. Using isentropic relations, the speed of 0.4 $U_\infty$ corresponds to a subsonic Mach number of about M = 0.6. These subsonic regions exist in the wake regions formed downstream of every pin. Finally, inside the wall boundary layer (Figure 11c) the spectra show the presence of only one branch, convecting at a speed slightly less than the freestream speed.

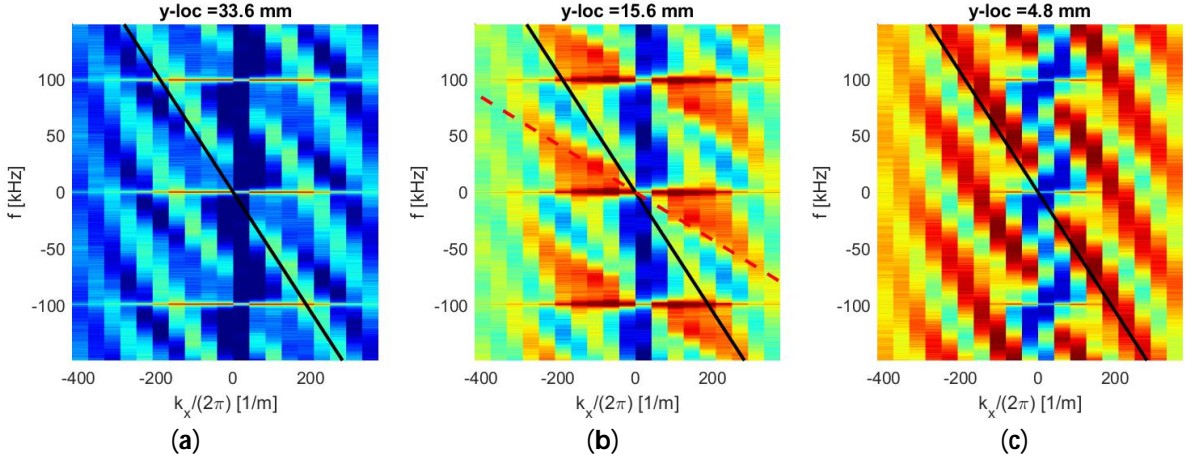

**Figure 11.** Stacked 2-D spectra of the deflection angles, plotted as $\log_{10}[S_\theta(f,k_x)]$, for (**a**) y = 33.6 mm, (**b**) y = 15.6 mm, and (**c**) y = 4.8 mm. The pin height is 25 mm. A solid black line indicates a convective speed of approximately $U_\infty$; a dashed red line in (**b**) denotes a convective speed of 0.4 $U_\infty$.

### 3.4. Streamwise Features behind Pins Visualized by Mie Scattering

The Mie scattering images in Figure 12 were taken using a 50 mm wide 532 nm laser sheet positioned on the centerline of the test section and against the upstream edge of the fused silica window. Similar to Figure 6, oblique and reflected shocks and a turbulent boundary layer are visible in Figure 12, as well as weak corner shocks in Figure 12b,c.

A notable feature in the images of Figure 12 is the low intensity of scattering directly downstream of the pin arrays. Since the stagnation temperature remains unchanged across the shocks, the lower Mach number flow directly behind the pin arrays results in increased static temperatures compared to the freestream, which prevents acetone recondensation directly behind the pin arrays. Due to the low flow velocity in the boundary layer and resulting higher static temperatures, Mie scattering is not present in the boundary layer immediately next to the solid wall. A region devoid of acetone droplets and thus Mie scattering is also present near the bottom wall of Figure 12a downstream of the reflected oblique shock due to the high gas temperatures in this region.

Two exceptions to this trend of droplet evaporation are in the shear layer downstream of the pin tips in Figure 12a and to a certain degree in Figure 12b, as well as in some structures in the turbulent shear layer near the upper wall. Since both the pin tip shear layer and the turbulent boundary layer on the upper wall must pass through the high-temperature lambda shock region upstream of the pin arrays, some portion of this flow must either experience rapid local expansion and the corresponding re-condensation of acetone droplets or ingest relatively cooler gas from another region in the flow.

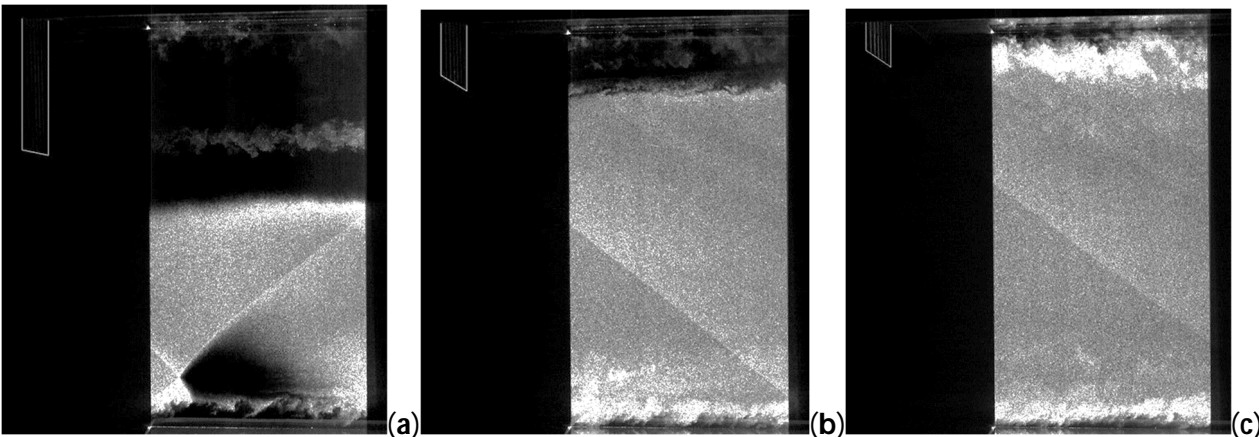

**Figure 12.** Single shot Mie scattering images from (**a**) 25 mm pins, (**b**) 10 mm pins, and (**c**) 5 mm pins. The pin array is outlined in white.

When the acetone concentration is increased to 6%, as shown in Figure 13, the shear layer at the pin tips and the boundary layer on the upper wall are more visible. A streamwise periodic structure is observed in the pin tip structures caused by the shedding of the vertical von Kármán vortices illustrated in Figure 1, and the boundary of the shear layer is visible as a bright horizontal strip. The high temperature region downstream of the reflected oblique shock seen in Figure 12 is also present in Figure 13 but is smaller in size, and a narrow dark line propagates downstream of the intersection of the separation and oblique shocks due to the localized high temperature at this point.

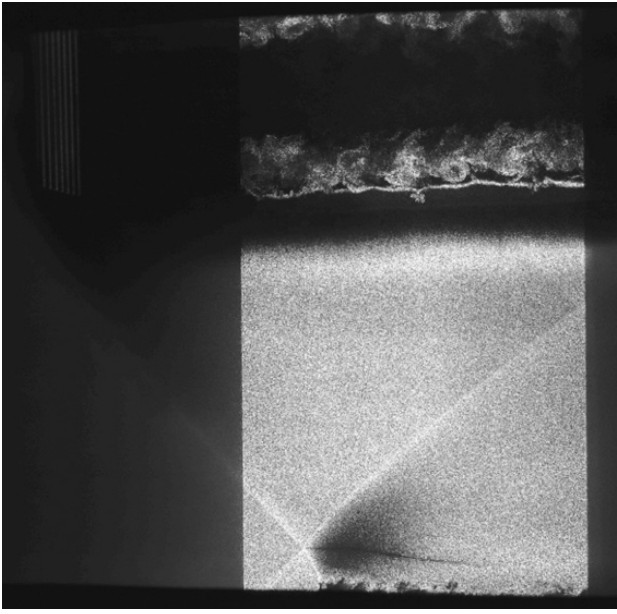

**Figure 13.** Instantaneous Mie scattering at 25 mm pins with 6% acetone vapors seeding and nonlinear brightness scaling.

*3.5. Cross-Flow Fine Scale Features Visualized by Mie Scattering*

Cross-flow Mie scattering images were taken using the same laser arrangement as for the streamwise images but with the cylindrical diverging lens rotated 90 degrees. Images were acquired through a fused silica window on the vacuum tank using a Nikon 200 mm f/4 lens and a Tokina 400 mm f/5.6 lens. The pins are visible in the images due to light scattering off their stainless-steel surfaces.

The oblique shock, seen as a diagonal line in Figure 12, is visible as a horizontal line in Figure 14 as it cuts through the plane of the image. The shock is not perfectly two-dimensional, and it curves slightly near the side walls. A turbulent boundary layer is visible on both the upper and lower walls, and large turbulent structures of increased brightness are present, which penetrate into the core flow. The physical mechanism responsible for the increased light scattering in these regions is discussed in Section 3.7 below. The nearly droplet-free region behind the pins in Figure 12 is also visible in Figure 14.

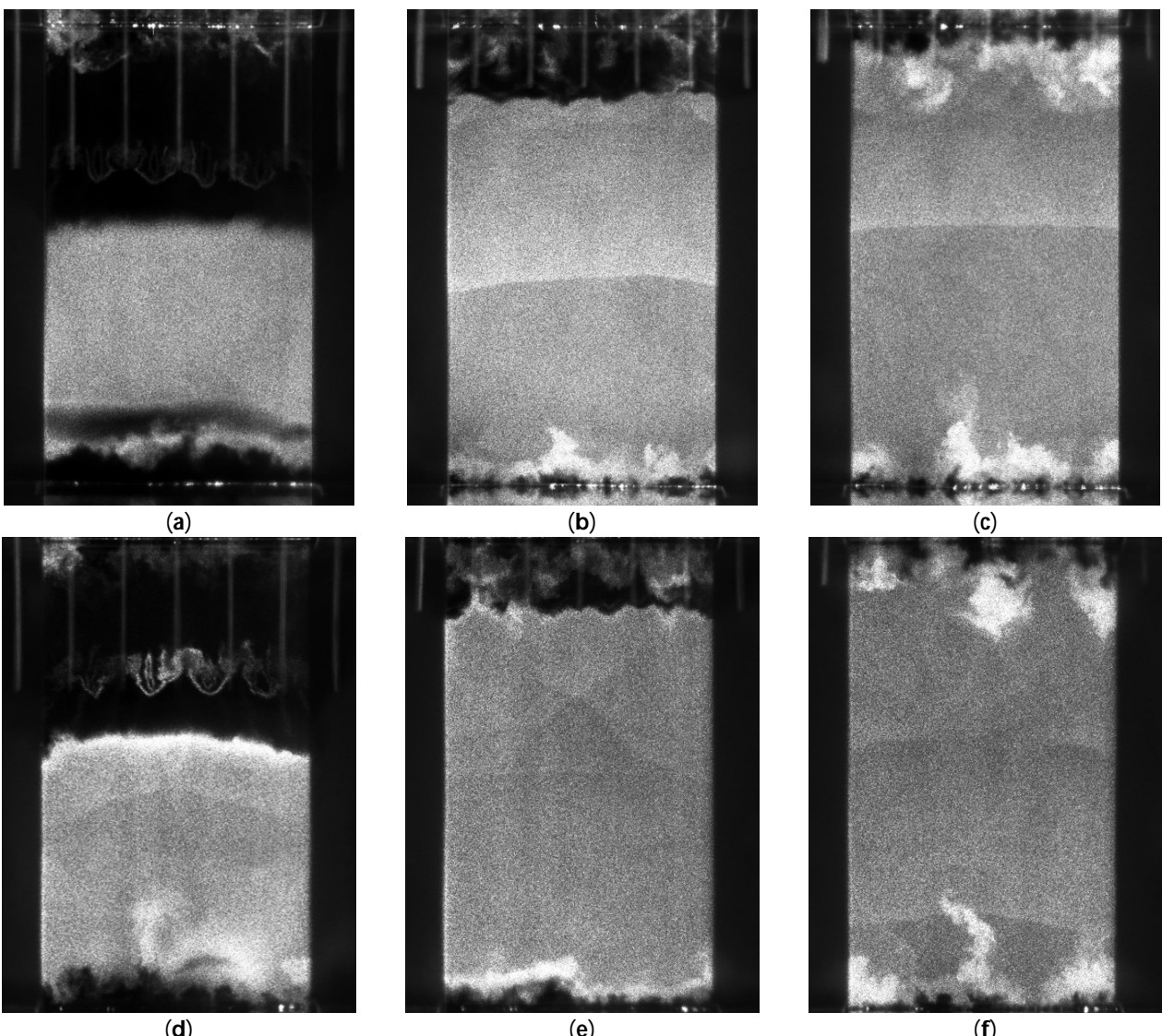

**Figure 14.** *Cont.*

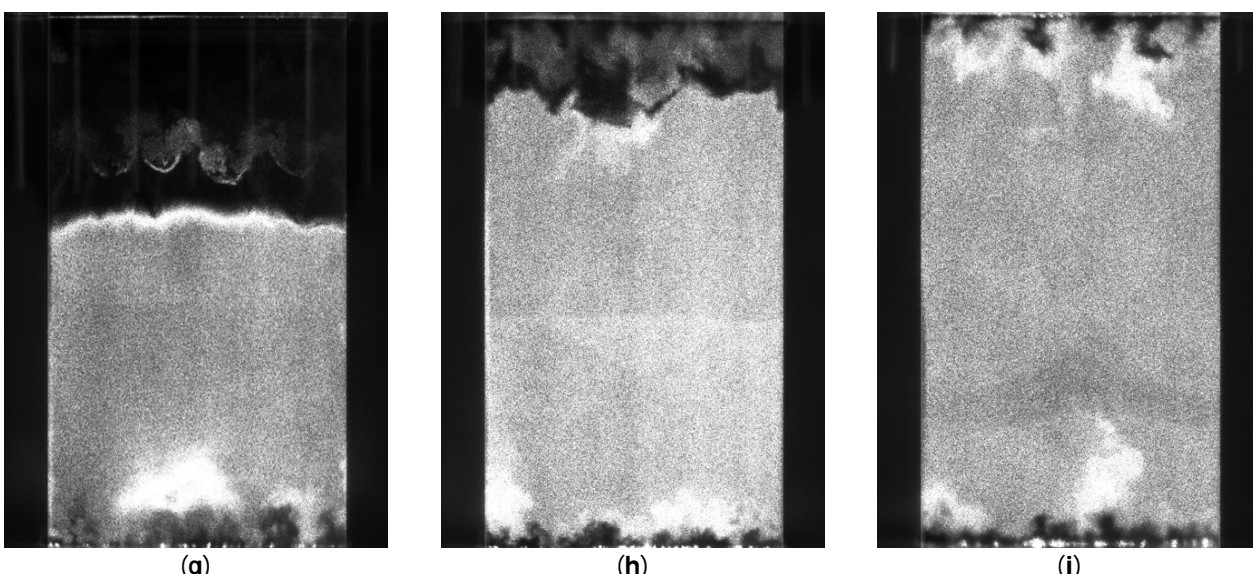

**Figure 14.** Mie scattering cross-flow single shot images. (**a**) 25 mm from 25 mm pins; (**b**) 25 mm from 10 mm pins; (**c**) 25 mm from 5 mm pins; (**d**) 50 mm from 25 mm pins; (**e**) 50 mm from 10 mm pins; (**f**) 50 mm from 50 mm pins; (**g**) 75 mm from 25 mm pins; (**h**) 75 mm from 10 mm pins; (**i**) 75 mm from 5 mm pins.

The droplet-free region behind the oblique shock impingement in Figure 12a is visible as a dark strip near the bottom of Figure 14a. The large-scale structure is visible in the interface between the warmer droplet-free region behind the pins and the cooler droplet-filled region in Figure 14, and the size of the structures is similar to the pin-to-pin spacing. These structures grow in size as the flow moves downstream and the two regions mix together.

Also visible in Figure 14 are fine-scaled structures near the end of the pin tips. These structures are especially evident for the 25 mm pins, which are shown in detail in Figure 15 at 25, 50, and 75 mm downstream with enhanced contrast.

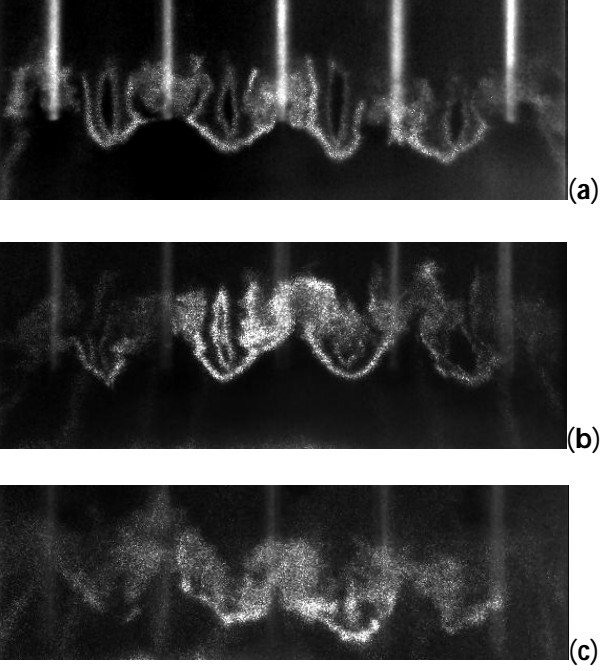

**Figure 15.** Pin tip flow structures (**a**) 25 mm, (**b**) 50 mm, and (**c**) 75 mm downstream of the 25 mm pins with enhanced contrast.

### 3.6. Visualization and Discussion of Different Flow Field Features

Several flow field areas are indicated in Figure 16, which are of particular interest for this work, including the fine-scaled structures near the end of the pin tips shown in Figures 15 and 16a. To the best knowledge of the authors, experimental visualization of such structures is not discussed in the available literature.

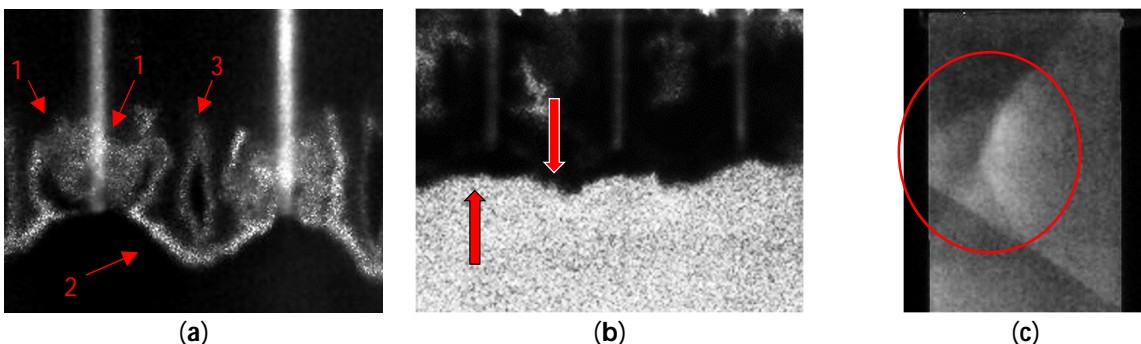

**Figure 16.** Flow features of special interest. (**a**) 25 mm pin tip structures (Mie scattering); (**b**) 10 mm pin array shear layer boundary (Mie scattering); (**c**) 25 mm pin array wake compression zone (PLIF).

The mechanism of formation of the pin tip structures is still not fully understood but may be reasonably hypothesized. In Figure 16a, structures (1) on either side of the 25 mm pin tip are believed to be pin-tip vortices similar to those illustrated in Figure 1a. The line (2) is the dividing surface between the accelerated flow between the pins and the flow below the pins, which passed through the curved bow shock and the expansion fan. The oval structure (3) is hypothesized to be caused by a jet of the relatively high pressure and high temperature gas just downstream of the curved bow shock near the pin tips into the lower pressure region between the pins caused by flow acceleration through the 2D nozzle created by adjacent pins. This inter-cylinder gas jet is cut off by the end of the expansion region. The interface of the flow between and below the pin tips is further distorted by the pin tip vortices and further gas expansion through the pin array. Once formed, this structure continues propagating downstream as a distinct structure visualized by Mie scattering due to temperature differences in the flow regions. This is schematically illustrated in Figure 17.

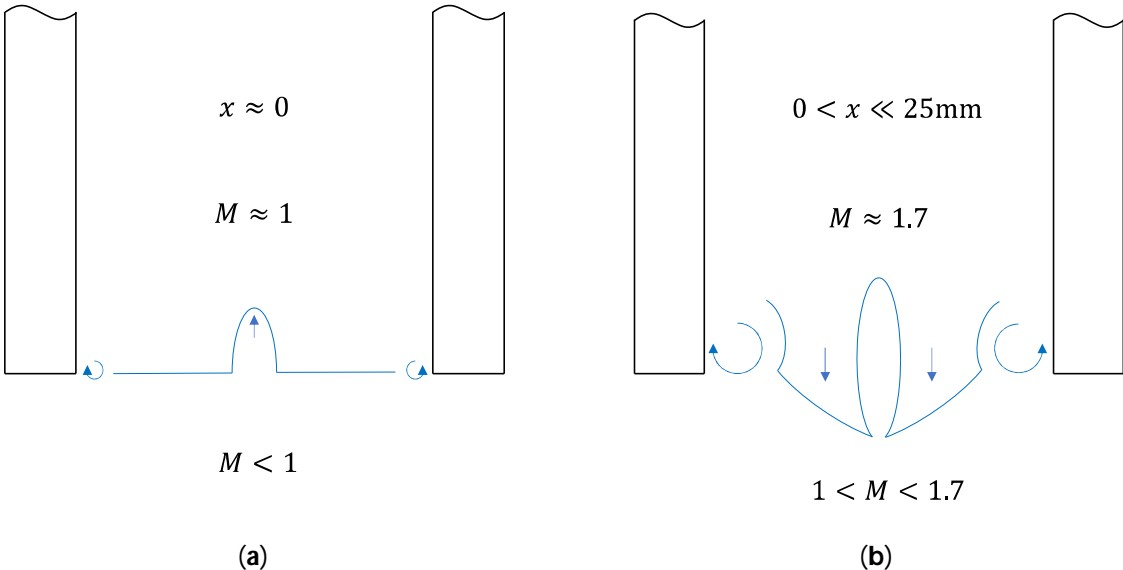

**Figure 17.** Hypothesized formation pattern of the observed pin tip structures (not to scale). (**a**) near the sonic region; (**b**) just downstream of the sonic region.

The shear layer boundary at the end of the 10 mm pins is visualized using Mie scattering in Figure 16b, and a regular pattern is observed in which the boundary is higher directly beneath the pins than between the pins. This is thought to be related to structure (2) in Figure 16a and due to flow expansion through the 2D nozzle created by adjacent pins.

The wake compression zone is visualized in Figure 16c by a single shot PLIF. This compression is caused first by the crossing of the reattachment and oblique shocks labeled in Figure 7, which occurs just upstream of the laser sheet. Crossing of the oblique shocks reflected off the side windows shown in Figure 9 results in increased compression at the center of the circled region in Figure 16c.

*3.7. Enhanced Scattering Regions*

Apparent in the Mie scattering in Figures 12–14 are regions of enhanced laser scattering, resulting in brighter regions of the flow. These regions are generally near an interface between warmer and cooler gas, such as the turbulent boundary layer in Figure 12c or at the boundary of the droplet-free region behind the pin tips in Figure 14g. The increased scattering in these areas is partially explained by local regions of increased gas density, and thus increased droplet density, in the turbulent boundary layer or immediately following a shock before droplet evaporation. In addition, since interfaces between warmer and cooler gas are regions with droplet evaporation and re-condensation, droplets in these regions are thought to be smaller and more numerous than in more equilibrated regions of the flow.

When the scattering plane is perpendicular to the direction of the electric vector of polarized light, the intensity of light scattered by a dielectric sphere [37] normalized by the incident light intensity is:

$$I = \frac{\lambda^2}{4\pi^2 r^2} |S|^2,$$

where $\lambda$ is the wavelength of the light, $r$ is the distance to the point of observation, and $S$ is the scattering amplitude function:

$$S = \sum_{n=1}^{\infty} \frac{2n+1}{n(n+1)} (a_n \pi_n \cos\theta + b_n \tau_n \cos\theta),$$

where $\theta$ is the angle of observation measured from the incident light direction. The angular functions are:

$$\pi_n = \frac{P_n^{(1)}(\cos\theta)}{\sin\theta}$$

and

$$\tau_n = \frac{d}{d\theta} P_n^{(1)}(\cos\theta),$$

where $P_n^{(1)}(\cos\theta)$ is the associated Legendre function of the first kind, and $a_n$ and $b_n$ are the coefficients given in [37]. For radii $a \lesssim \lambda/10$, the Rayleigh approximation for the scattering amplitude is:

$$S = \left(\frac{2\pi a}{\lambda}\right)^3 \left(\frac{m^2-1}{m^2+2}\right),$$

where $m$ is the relative refractive index, which is approximately $m = 1.36$ for acetone.

As shown in Figure 18, the magnitude of the scattering amplitude function squared $|S|^2$ at $\theta = \pi/2$ scaled by the droplet volume $v = (4/3)\pi a^3$ has a maximum at a radius of about $a = 130$ nm for $\lambda = 532$ nm light. Thus, while scattering from a spherical droplet increases with increasing droplet radius, the scattered light intensity per unit volume of droplets has a maximum at a droplet radius of approximately $a = \lambda/4$. This results in increased light scattering in regions with numerous small droplets. The linear region at small $a$ in Figure 18 corresponds to the Rayleigh scattering regime.

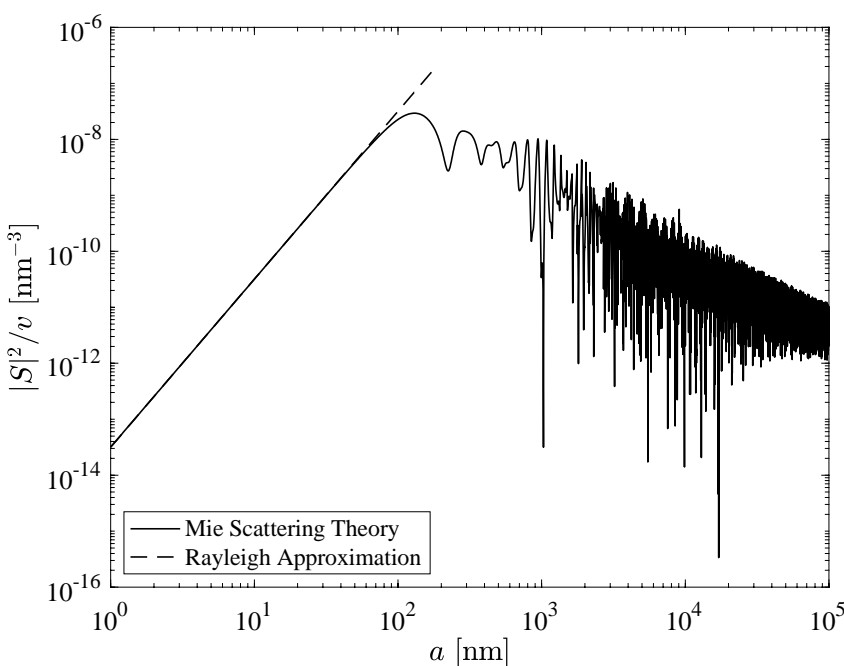

**Figure 18.** Mie scattering amplitude scaled by droplet volume for $\lambda = 532$ nm light.

## 4. Conclusions

Three-dimension flow fields behind cylindrical wall-normal pin arrays of various heights have been studied using several optical methods, including schlieren imaging, acetone PLIF, acetone Mie scattering, and wavefront measurements using a Shack–Hartmann wavefront sensor. These methods have been used to identify various flow structures such as the lambda shock region, reattachment shocks, shock wave impingements, and the supersonic shear layer originating from the pin tips. It should be noted that a successful visualization of some flow features is possible because the SBR-50 facility at the University of Notre Dame allows for an accurate adjustment of the gas temperature, which makes Mie scattering visualization an appropriate method for the recognition of fine-scale structures in the mixing layer behind the pins.

Using different methods, the present study examined the dynamics of a complex compressible flow. Particularly, the regular structure of the supersonic shear layer generated by the pin tips was captured. This has not previously been identified and discussed. On the other hand, detailed visualization of the fine structure of the flow field demonstrated that the Mie scattering method is a powerful tool for the reconstruction of flow field details when the gas density gradients are too low to be well-detectable with commonly used techniques such as schlieren, aero-optical distortions detection, or even laser differential interferometry (LDI). In terms of sensitivity, this method is especially beneficial when the flow temperature is accurately adjustable.

Most features of the flow field acquired with these optical methods are well predictable. However, the description of a few peculiarities requires further analysis. Among them is the structure of the mixing layer behind the pin array where the pin-activated vorticity and flow acceleration through the 2D nozzle formed by adjacent pins leads to the generation of complex flow structures in the supersonic zone as visualized by Mie scattering. Such an analysis will likely require a quantitative three-dimensional velocimetry technique, such as stereo particle image velocimetry (PIV) or a more advanced laser-based method, in close proximity to the pin tips. Another focus of upcoming efforts is a quantitative analysis of fast schlieren records and Shack–Hartmann datasets to reveal details of the flow velocity distribution behind the pin array. As an important practical task, an in-depth study of the regular flow perturbations behind the pins array is essential. It is also important to examine the relevant optical distortions associated with light propagation across the shear layer. All

of these should be combined with a CFD numerical analysis as is indicated in the objectives' statement in Section 1.

**Author Contributions:** Conceptualization, S.G., M.R.K. and S.B.L.; methodology, S.G. and S.B.L.; investigation, P.A.L. and S.E.; resources, M.R.K. and S.G.; writing—original draft preparation, P.A.L., S.G. and S.B.L.; writing—review and editing, S.B.L. and S.G.; visualization, P.A.L. and S.E.; supervision, S.G. and S.B.L.; project administration, S.G. and M.R.K.; funding acquisition, M.R.K. All authors have read and agreed to the published version of the manuscript.

**Funding:** This research was funded by the US Air Force Research Laboratory, Cooperative agreement number FA9451-17-2-0088. The U.S. Government is authorized to reproduce and distribute reprints for governmental purposes notwithstanding any copyright notation thereon. Approved for public release; distribution is unlimited. Public Affairs release approval #AFRL-2022-2716.

**Data Availability Statement:** The data are available upon request from the corresponding author.

**Acknowledgments:** The authors would like to thank Matthew Orcutt for providing help in setting up the optical system to collect the wavefronts.

**Conflicts of Interest:** The authors declare no conflicts of interest.

## Nomenclature

| | |
|---|---|
| $a$ | droplet radius |
| $d$ | pin diameter |
| $h$ | pin height |
| $l$ | pin center-to-center distance |
| $S$ | scattering amplitude function |
| $v$ | droplet volume |
| $\gamma$ | specific heat ratio |
| $\theta$ | shock wave angle *or* angle from incident light direction |
| $\rho$ | gas mass density |

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
