# Peer review of "Flow Structure behind Spanwise Pin Array in Supersonic Flow"

_aerospace, doi:10.3390/aerospace11010093_

Round 1
Reviewer 1 Report
Comments and Suggestions for Authors
In the paper under consideration the experimental results on 3Dvisualization of a flow structure behind an array of cylindrical pins installed on a wall of a supersonic duct are presented. This geometry may simulate several instruments, such as a supersonic mixer, a turbulence-generating grid, or a grid fin. Authors use fast schlieren visualization, Shack-Hartmann wavefront sensor, acetone Mie scattering, and acetone planar laser-induced fluorescence. Note, that the Mie scattering technique presented by the authors has not been previously described in the literature.
The article contains new interesting results that may be of practical importance. The presented results are distinguished by their high resolution of flow details such as shock wave structures, shear layers, boundary layers and vortex lines. The paper undoubtedly can be published in Aerospace.
Reviewer has only one suggestion: In Fig. 5 it should be noted the visualization of boundary layers.
Author Response
The authors wish to express their thanks to the reviewers whose comments and patience helped revise and finish the manuscript. The changes made reflect the authors’ suggestions for this manuscript in response to the reviewer’s comments and are highlighted in an accompanying file.
Reviewer has only one suggestion: In Fig. 5 it should be noted the visualization of boundary layers.
We added the BL indication to Fig.5 and a short phrase to the text.
Again, the authors appreciate the work done by, suggestions provided by, and patience of the reviewer. We hope that our modifications and responses to comments are satisfactory.

Reviewer 2 Report
Comments and Suggestions for Authors
In this manuscript, authors attempted to experimentally study the supersonic flow structure behind the arrays of cylinderical pins used in walls of supersonic duct. Authors used various non-intrusive flow diagnostics techniques such as high speed Schlieren imaging, Shack-Hartmann wavefront sensor, acetone Mie scattering, and acetone planar laser-induced fluorescence to reconstruct a three-dimensional portrayal of the flow interaction with the pin array. These optical diagnostics allowed to visualize and identify the shock layer, separation zone, mixing zone and triple point etc., in the supersonic flow.
I appreciate the authors for studying complex flow structures using various non-intrusives techniques which are interesting and important for validation of numerical simulations. Author’s discussions on the flow physics are appreciable. I have few queries and comments on this manuscript, which are given below,
1. Abstract written such a way that the flow features are emphasized and flow physics are discussed more in the manuscript which true. But the conclusion written as if the manuscript emphasis only diagnostics techniques with its capabilities and limitations. I suggest authors to give little attention to that part. Otherwise the content is good.
2. Schematic for experimental setup is not quiet clear. I suggest authors to show experimental setup diagram in 3D diagram and explain about it, so that it’s easier for readers.
3. Are these imaging done simultaneously?
4. It is not clear the images presented in the result and discussion part are instantaneous images or mean images (averaged in time).
Author Response
The authors wish to express their thanks to the reviewers whose comments and patience helped revise and finish the manuscript. The changes made reflect the authors’ suggestions for this manuscript in response to the reviewer’s comments and are highlighted in an accompanying file.
- Abstract written such a way that the flow features are emphasized and flow physics are discussed more in the manuscript which true. But the conclusion written as if the manuscript emphasis only diagnostics techniques with its capabilities and limitations. I suggest authors to give little attention to that part. Otherwise the content is good.
Answer 1. We made a modification of the Conclusions shifting focus to the flow physics.
- Schematic for experimental setup is not quiet clear. I suggest authors to show experimental setup diagram in 3D diagram and explain about it, so that it’s easier for readers.
Answer 2. We revised Fig.2 and clarified the related description.
- Are these imaging done simultaneously?
Answer 3. The imaging was performed under identical conditions but not simultaneously. For example, Mie scattering and PLIF visualization require the flow seeding and laser illumination which conflicts with the schlieren and SH sensing.
- It is not clear the images presented in the result and discussion part are instantaneous images or mean images (averaged in time).
Answer 4. The most images are instantaneous except for Fig.4 and Fig.10 where schlieren image illustrate the geometry. We added the comments when missed.
Again, the authors appreciate the work done by, suggestions provided by, and patience of the reviewer. We hope that our modifications and responses to comments are satisfactory.
